# Genome-Wide Identification of *MYC* Transcription Factors and Their Potential Functions in the Growth and Development Regulation of Tree Peony (*Paeonia suffruticosa*)

**DOI:** 10.3390/plants13030437

**Published:** 2024-02-02

**Authors:** Qianqian Wang, Bole Li, Zefeng Qiu, Zeyun Lu, Ziying Hang, Fan Wu, Xia Chen, Xiangtao Zhu

**Affiliations:** College of Jiyang, Zhejiang A&F University, Zhuji 311800, China; qianqianwang@stu.zafu.edu.cn (Q.W.); bolerlee0128@gmail.com (B.L.); c979923111@outlook.com (Z.Q.); wdmmtl7@163.com (Z.L.); hzy123452024@163.com (Z.H.); 13173755206@163.com (F.W.)

**Keywords:** *Paeonia suffruticosa*, *MYC* transcription factor, growth and development, hormone signaling, abiotic stress

## Abstract

Tree peony (*Paeonia suffruticosa* Andr.) is a traditional Chinese flower with significant ornamental and medicinal value. Its growth and development process is regulated by some internal and external factors, and the related regulatory mechanism is largely unknown. Myelocytomatosis transcription factors (*MYCs*) play significant roles in various processes such as plant growth and development, the phytohormone response, and the stress response. As the identification and understanding of the MYC family in tree peony remains limited, this study aimed to address this gap by identifying a total of 15 *PsMYCs* in tree peony and categorizing them into six subgroups based on bioinformatics methods. Furthermore, the gene structure, conservative domains, *cis*-elements, and expression patterns of the *PsMYCs* were thoroughly analyzed to provide a comprehensive overview of their characteristics. An analysis in terms of gene structure and conserved motif composition suggested that each subtribe had similarities in function. An analysis of the promoter sequence revealed the presence of numerous *cis*-elements associated with plant growth and development, the hormone response, and the stress response. qRT-PCR results and the protein interaction network further demonstrated the potential functions of *PsMYCs* in the growth and development process. While in comparison to the control, only *PsMYC2* exhibited a statistically significant variation in expression levels in response to exogenous hormone treatments and abiotic stress. A promoter activity analysis of *PsMYC2* revealed its sensitivity to Flu and high temperatures, but exhibited no discernible difference under exogenous GA treatment. These findings help establish a basis for comprehending the molecular mechanism by which *PsMYCs* regulate the growth and development of tree peony.

## 1. Introduction

The basic helix-loop-helix family (bHLH) is a prominent gene family in plants, playing crucial roles in plant growth, secondary metabolism, and signal transduction [1,2]. As constituents of the bHLH superfamily, the Myelocytomatosis transcription factors (*MYCs*) contain a highly conserved HLH domain and bHLH-MYC-N domain [3]. *MYCs* were initially discovered in *Zea mays* and have since been extensively identified in various plant species, such as *Arabidopsis thaliana*, *Oryza sativa*, and *Triticum aestivum*, among others [4,5].

Functional studies on *MYCs* in various plant species have revealed their involvement in the plant hormone response, stress response, and growth and development processes [2,6,7]. In *Arabidopsis*, specifically, *AtMYC2*, *AtMYC3*, and *AtMYC4* have been identified as key players in the Jasmonate (JA) signaling pathway, exerting regulatory control over plant development, seed production, and secondary metabolism accumulation [2,3]. Furthermore, *AtMYCs* were found to play crucial roles in the regulation of responses mediated by Ethylene, Gibberellin (GA), and Abscisic acid (ABA) [8,9]. In *Marchantia polymorpha*, *MpMYCs* exhibit similar functionalities as in *AtMYC2*, including nuclear localization, interactions with JA repressors, and regulation by light [5]. However, unlike their *Arabidopsis* orthologs, *MpMYCs* do not play a role in regulating fertility. In *T. aestivum*, *O. sativa* and *B. distachyon*, the *MYC* homologous genes primarily function in the plant’s growth, development, and stress response [10]. Similar findings were observed in *Zea mays*, where the expression of *ZmMYCs* was significantly increased under drought stress conditions [11].

Tree peony (*Paeonia suffruticosa* Andrews), renowned as one of the top ten famous flowers in China, holds significant importance as a horticultural plant due to its ornamental, nutritional, and medicinal values [12,13]. Previous studies showed that the growth and development process of tree peony is regulated by some internal and external factors [14]. In field conditions, the primary constraint on bud burst and flowering lies in the release of bud endodormancy. Artificial chilling and the application of phytohormones, such as gibberellins (GAs), garlic paste, 5-azacytidine (5-azaC), and ABA, were considered as effective strategies [15,16,17]. Additionally, several crucial factors, including *TARGET OF EAT* (*PsTOE3*), β-1,3-glucanase gene *PsBG6*, *PsMYB1*, and others, have been confirmed to be involved in the dormancy release process [15,18]. Besides that, during the growth and development of tree peony, high temperatures and heat injuries are also significant obstacles [19]. They can disrupt the photosynthetic mechanism and influence the function of PSII and the physiological characteristics of tree peony [13]. And some studies have initially explored the mechanisms underlying the response to heat stress in tree peony [20]. Although recent academic interest has focused on investigating the growth and development regulation as well as stress response, the related regulatory mechanism is largely unknown [13,14,15].

The MYC family plays significant roles in various plant processes. While in tree peony, public information is lacking, and the specific role of *PsMYCs* in the growth and development process remains largely unexplored. In the present study, 15 *PsMYC* genes were identified from the genomes of tree peony. Their classification, gene structure, motif composition, chromosomal distribution, evolution, and *cis*-regulatory elements (CREs) were analyzed. The expression levels of *PsMYC* genes in various tissues and developmental stages were quantified. Besides that, the expression pattern and promoter activity of important *PsMYCs* under different exogenous hormone treatments and high-temperature stress were analyzed. This study aimed to support a basis for further investigations on the functions of *PsMYCs* and provide a reference for revealing the molecular mechanisms of the growth and development process in tree peony.

## 2. Materials and Methods

### 2.1. Plant Materials and Treatment Methods

*P. suffruticosa* ‘Lu He Hong’ was used as plant material here and grown at Jiyang College of Zhejiang A&F University (Zhejiang, China, 29°75′52″ N, 120°26′12″ E). All plant tissue materials for expression analysis were collected from 5-year-old trees, which had entered the flowering age and grown well under natural conditions, and stored at −80 °C.

For exogenous hormone treatments, the dormant buds of tree peony were sprayed with an aqueous solution containing 0.1% (*v/v*) phosphoric acid, 0.025% (*v/v*) Triton X-100, and GA_3_ or Fluridone with 300 mg/L (Flu, a synthesis inhibitor of ABA) (Dingguo Biotechnology Co., Ltd., Guangzhou, China). The control plants were sprayed with a solution containing only 0.1% (*v/v*) phosphoric acid and 0.025% (*v/v*) Triton X-100. All plants were cultivated for 14 h/10 h of light/dark at 25 °C, and the materials were collected 7 days after treatment. For the high-temperature treatment, the plants were cultivated at 25 °C (for the control) and 40 °C (for the study group), and the samples were collected 2 days after treatment.

### 2.2. Identification of the MYC Gene Family in Tree Peony

The genome sequence data and the annotation information of tree peony (*P. ostii* ‘Fengdan’) were obtained from the China National Gene Bank database (https://ftp.cngb.org/pub/CNSA/data5/CNP0003098/CNS0560369/CNA0050666/, accessed on 12 September 2023) [21]. The MYC protein of tree peony was identified with both the bHLH domain and the specific MYC domains. The candidate protein sequences were subjected to the online domain analysis program NCBI-CDD 1.0 (https://www.ncbi.nlm.nih.gov/cdd/, accessed on 12 September 2023) and SMART 8.0 (http://smart.emblheidelberg.de/, accessed on 12 September 2023) to confirm the conserved domains. The MYC protein of *A. thaliana* (At) and *O. sativa* (Os) were downloaded from TAIR (http://www.arabidopsis.org/, accessed on 12 September 2023) and TIGR (http://www.tigr.org/, accessed on 12 September 2023). The physical and chemical characteristics of the MYC proteins were analyzed using online software Expasy 3.0 (http://web.expasy.org/protparam/, accessed on 12 September 2023). The neighbor-joining phylogenetic tree of protein from different species was constructed with 1000 bootstrap replicates using MEGA 8.0 software.

### 2.3. Sequence Structure, Conserved Motifs, and Chromosomal Location Analysis

The gene structures of the identified *PsMYCs* were analyzed using the online software GSDS 2.0 (http://gsds.gao-lab.org/ (accessed on 2 November 2023)). The motifs of the *PsMYCs* were analyzed using the MEME Suite 5.3.0 at a maximum motif number of 10, with a minimum and maximum width of 6 and 50, respectively (http://memesuite.org/tools/meme (accessed on 2 November 2023)). The positional information of the *PsMYC* genes was collected based on the genome annotation information. A visualization of the chromosomal locations of the *PsMYCs* was carried out using Circos-0.69 6.0 (http://circos.ca/, accessed on 2 November 2023).

### 2.4. Prediction of the Cis-Elements in the Promoter of PsMYCs and Promoter Cloning

The upstream sequences (2 kb) of the *PsMYC* coding sequences were obtained from the genome database of *P. ostii* ‘Fengdan’. The online software PlantCARE was used to analyze the *cis*-acting regulatory elements in the promoter of the *PsMYCs* (http://bioinformatics.psb.ugent.be/webtools/plantcare/html/ (accessed on 2 November 2023)). A heat map of the *cis*-acting regulatory elements was drawn with the GraphPad prism 8 software.

DNA was extracted from the leaves of ‘Lu He Hong’ using a DNA extraction kit (Vazyme, Nanjing, China), and the methods used referred to the manufacturer’s instructions. Based on the transcriptome and genome of tree peony, the primers of the *PsMYC2* promoter were designed using Primer Premier 5 software.

### 2.5. RNA Extraction and Expression Analysis

Total RNA was extracted using the RNAprep Pure Plant Kit (TianGen, Beijing, China), and the quality was detected using a nucleic acid analyzer (Implen Company in Germany). First-strand cDNA was synthesized using a PrimeScriptTM RT reagent Kit (TaKaRa, Dalian, China).

The expressions of related genes were detected via quantitative real-time polymerase chain reaction (qRT-PCR) on a Light Cycller 480II Real Time PCR system (Roche, Basel, Switzerland). Primer Premier 5 software was used to design the primers for the qRT-PCR reactions, and *PsACT* was used as the reference gene [12]. The reaction system was as follows: SYBR Premix Ex Taq 10 μL, cDNA 2 μL, upstream and downstream primer (10 μmol/L), 8 μL each, and ddH_2_O supplemented to 20 μL. The reaction procedure was as follows: 95 °C for 30 s, 95 °C for 5 s, 60 °C for 30 s, a total of 40 cycles; 95 °C for 5 s, 60 °C for 1 min, 95 °C for 15 s, three biological replicates. The relative expressions were calculated using the 2^−△△CT^ method [22,23].

### 2.6. Luciferase Assay

To investigate the regulation of *PsMYC2* by different plant hormones and high temperature, a Luciferase assay was performed. The promoter of *PsMYC2* was cloned into pGreen0800II-LUC, and the vectors with *PsMYC2*pro::LUC and pGreen0800II-LUC were transformed into GV3101 (including pSoup-p19). *PsMYC2*pro::LUC and pGreen0800II-LUC were transiently transformed into the leaves of *Nicotiana tabacum*. After that, the *N. benthamiana* was cultivated in the dark for 24 h at 25 °C, and then the different treatments were performed. The activities of fluorescein enzymes were measured after the treatments. The activities of firefly luciferase (LUC) and renilla luciferase (REN) were measured through the GLOMAX^®^ multifunctional instrument (Promega, Madison, WI, USA).

### 2.7. The Protein Interaction Network Predicition

In order to enhance comprehension on the potential interactions between PsMYCs and other proteins, the online software SRING v11.5 (https://cn.string-db.org/, accessed on 2 November 2023) was employed to construct a protein interaction network map. The method and organism selected were the “amino acid sequence of PsMYCs and “*Arabidopsis*”, respectively, while default parameters were utilized for the remaining aspects. Ultimately, the protein interaction network map was created by selecting the most similar *A. thaliana* proteins based on the bit score and E-value.

## 3. Results

### 3.1. Identification of PsMYC Genes in Tree Peony

Genes homologous to the MYC family in tree peony were deduced on the basis of the transcriptome and genome data, and a total of 15 putative *PsMYC* genes were identified, named *PsMYC1–15*. Then, we investigated the physical and chemical characteristics of 15 *PsMYC* genes. The results show that the coding sequence lengths of 15 *PsMYCs* ranged from 1071 (*PsMYC9*) to 2850 bp (*PsMYC7*), and the lengths of amino acids ranged from 357 to 950 (Appendix A). The molecular weights (MWs) varied from 40.62 kDa (*PsMYC9*) to 103.92 kDa (*PsMYC7*), and the isoelectric points (IPs) changed from 4.65 (*PsMYC11*) to 7.62 (*PsMYC13*). Besides that, the analysis results of the instability index (II) and GRAVY showed that most of the PsMYC proteins were unstable and hydrophilic (Appendix A).

### 3.2. Phylogenetic Tree of PsMYCs

To reveal the evolutionary relationships of the *MYC* genes among *A. thaliana*, *O. sativa*, and tree peony, a neighbor-joining phylogenetic tree was constructed based on 16 *AtMYCs* from *A. thaliana*, 7 *OsMYCs* from *O. sativa*, and 15 *PsMYCs* from tree peony. The results show that 15 *PsMYCs* were clustered into six sub-groups: *PsMYC8* and *PsMYC12* within group 1; *PsMYC4* and *PsMYC11* within group 2; only *PsMYC6* within group 3; five *PsMYCs* (*PsMYC1/9/13/14/15*) within group 4; *PsMYC3*, *PsMYC5*, and *PsMYC7* within group 5; and *PsMYC2* and *PsMYC10* within group 6 (Figure 1). The 15 *PsMYCs* distributed into different groups indicated the diversity in structure and function.

### 3.3. Gene Structures, Conserved Motifs, and Chromosomal Location Analysis of PsMYCs

The Gene Structure Display Server (GSDS) was used to investigate the exon–intron distribution of the 15 *PsMYC* genes (Figure 2A). The results suggest that the presence and number of introns in the *PsMYCs* were different, and the number ranged widely within 0–11. There was no introns in 4 *PsMYCs*, and more than 5 introns in 9 *PsMYCs* (Figure 2A). The sequence diversity in the number of introns from the 15 *PsMYCs* indicated that the *PsMYCs* might have experienced extensive domain shuffling after genome duplication.

Through MEME 8.0 software, 10 conserved motifs were identified among the 15 *PsMYCs* (Figure 2B and Appendix A). The results indicate that the conserved motif number in the PsMYC proteins varied widely and ranged from 3 to 8, in which motif 5 was highly conserved in all proteins. In addition, motifs 2, 6, 4, and 7 were relatively conserved in the PsMYC proteins during evolution in the tree peony (Figure 2B). The analysis of chromosomal locations revealed that four *PsMYCs* (*PsMYC2/4/5/9*) were situated on chr01, four *PsMYCs* (*PsMYC1/7/13/15*) were positioned on chr02, five *PsMYCs* (*PsMYC3/6/8/11/12*) were found on chr03, and *PsMYC14* was localized on chr04 (Appendix A).

### 3.4. Promoter Analysis of PsMYC Genes

To understand the *cis*-regulatory elements (CREs) in the promoter of *PsMYCs*, 2 kb upstream sequences before the transcription initiation site were identified and analyzed using the PlantCARE web tool. It showed that a large number of the CREs related to the light, plant growth and development, hormones, stress, and different transcription factor binding sites were present in the promoter of the 15 *PsMYCs* (Figure 3A). Among the CREs, we focused on the phytohormone response, stress response, and growth and development-related types. As shown in Figure 3B, Jasmonate acid (JA)- and Abscisic acid (ABA)-responsive elements existed in a majority of the *PsMYCs*, implying that the expression of *PsMYCs* might be regulated by JA and ABA. Regarding the stress response, the majority of *PsMYCs* exhibited anoxic-induction elements (ARE), stress responsive elements (STRE), and drought-inducibility element (MBS), suggesting their potential significance in the drought stress response. In terms of growth and development, G-box elements were found in most *PsMYCs*, and six *PsMYCs* contained a circadian motif, which are implicated in photoperiodic regulation (Figure 3B). These results suggest that the 15 *PsMYCs* may be involved in various biological processes in tree peony.

### 3.5. Tissue-Specific Expression Patterns of PsMYCs

In order to investigate the expression patterns of the 15 *PsMYCs* in various tissues, including the root, stem, leaf, dormant bud, and dormancy release bud, a qRT-PCR analysis was performed. As shown in Figure 4, all *PsMYCs* exhibited significantly lower expression levels in both root and stem tissues, and only *PsMYC1* and *PsMYC8* displayed a low accumulation in leaf. Notably, distinct expression patterns were observed between the dormant bud and dormancy release bud. Five genes (*PsMYC1/4/5/11/13*) exhibited higher expression levels in the dormancy release bud compared to the dormant bud. Conversely, three genes (*PsMYC2/7/8*) demonstrated lower expressions in the dormancy release bud relative to the dormant bud (Figure 4). These results indicate that *PsMYCs* may display different roles in the growth and development processes of tree peony.

### 3.6. Expression Patterns of PsMYCs in the Growth and Development Process

To further study the expression patterns of *PsMYCs*, their expression levels during the growth and development processes were analyzed, including bud dormancy stage (S1), bud dormancy release stage (S2), blooming bud stage (S3), wind bell stage (S4), and full flowering stage (S5). This showed that most *PsMYCs* (except *PsMYC1)* had similar expression patterns and were significantly upregulated from S4 to S5, suggesting that those genes may have important roles in the flower opening process (Figure 5). Besides that, contrasting expression patterns between *PsMYC1* and *PsMYC2* were revealed. Specifically, *PsMYC1* exhibited an increase in expression during bud dormancy release, followed by a subsequent decrease during later growth and development, while *PsMYC2* was downregulated during bud dormancy release but displayed a noticeable upregulation from S2 to S5, suggesting its potential inhibitory effect on bud dormancy release and promotive effect on later growth and development. In addition, the expression patterns of *PsMYC9* and *PsMYC10* were similar to that of *PsMYC2* (Figure 5). These findings show that the *PsMYC* genes might possess distinct regulatory roles in the growth and development processes and that *PsMYC1/2/9/10* deserves more attention.

### 3.7. Expression Patterns of PsMYC1/2/9/10 under Different Treatments

In order to reveal the potential functions of *PsMYC1/2/9/10* in the growth and development processes, we analyzed their expression patterns under different treatments, including GA, Flu, and high temperature (Figure 6). The results show that compared with the control, *PsMYC1/9/10* showed no significant differences in expression levels, while only the expression of *PsMYC2* was obviously regulated by different treatments. It was found that the accumulation of *PsMYC2* was decreased under the GA and Flu treatments and increased under high-temperature stress, suggesting the important role of *PsMYC2* in growth, development, and the high-temperature stress response.

### 3.8. Effects of Different Treatments on the Promoter Activity of PsMYC2

The *PsMYC2* promoter was inserted into a pGreenII 0800-Luc vector, and a transient luciferase assay was performed to investigate the promoter activity under different treatments (Figure 7). The results show that the luciferase signal intensity of *PsMYC2*pro::Luc was decreased under exogenous Flu treatment. However, compared to the control (CK), the signal intensity had no obvious difference under the exogenous GA_3_ treatment. These results indicate the important role of *PsMYC2* in an ABA-mediated growth and development process. In addition, the promoter activity of *PsMYC2* was obviously activated by high temperature, suggesting the role of *PsMYC2* in the high-temperature stress response.

### 3.9. The Interaction Network Prediction of PsMYCs

To investigate the regulatory pathways involving PsMYC proteins and other proteins, a network map was constructed using the STRING website (Figure 8). Among the 15 PsMYC proteins, PsMYC2 exhibited a structural similarity to the *A. thaliana* protein AtMYC2 and demonstrated interactions with PsMYC4 and PsMYC3/5 proteins (Figure 8A). Additionally, PsMYC2 displayed interactions with various other proteins, such as MYB, TIFY, and RGA (Figure 8B). These findings suggest that PsMYC2 not only forms heterodimers to exert its function in tree peony, but also engages in interactions with other proteins to influence the growth and development processes of the plant.

## 4. Discussion

The plant growth and development process is regulated by complex gene networks, and the MYC gene family is an important player [2,24]. It has been widely identified in plants, while the number of the MYC gene family members varies in different species. In *T. aestivum*, *O. sativa*, *B. distachyon*, and *Saccharum spontaneum*, 26 *TaMYC*, 7 *OsMYC*, 7 *BdMYC*, and 23 *SsMYC* genes have been identified, respectively, and their features have been characterized [10,25]. However, there are few reports regarding the study of *MYCs* in tree peony. In this study, a total of 15 *PsMYC* genes were identified and clustered into six sub-groups based on phylogenetic analysis with the *MYC* genes of *A. thaliana* and *O. sativa* (Figure 1).

Besides that, the expression profiles of *PsMYCs* were analyzed. Most *PsMYC* genes were highly accumulated in the leaves, dormant bud, and dormancy release bud, and differentially expressed during the growth and development process of tree peony (Figure 4 and Figure 5). The combination of their *cis*-elements related to plant growth and development, phytohormone responsiveness, and stress responsiveness further suggests their potential functions In the growth and development process (Figure 3). In *T. aestivum*, *O. sativa*, and *B. distachyon*, most *MYC* genes are expressed in roots, stems, leaves, and inflorescences, and mainly function in growth and development [10]. The *MYC* genes in *A. thaliana* were found to play significant roles in various pathways, such as primary root growth, anthocyanin biosynthesis, oxidative stress tolerance, light-mediated photomorphogenic growth, resistance to necrotrophic fungi, and the biosynthesis of tryptophan and indoleglucosinolates [2,3]. Similarly, *HbMYCs* have been identified as crucial factors in the differentiation and biosynthesis processes of *Hevea brasiliensis* [26]. In *Artemisia annua*, *AaMYC2* has been identified to activate the expression of *CYP71AV1* and *DBR2*, resulting in the promotion of artemisinin biosynthesis [27].

In tree peony, bud dormancy release is a critical stage in the life cycle, directly influencing growth, development, and flowering [28]. Besides chilling, GAs and ABA or its inhibitor Flu are fundamental phytohormones that extensively regulate plant growth and development, especially bud dormancy release [16,29]. In this study, based on the expression patterns of the *PsMYCs* during the growth and development process of tree peony (S1–S5), we paid more attention to *PsMYC1/2/9/10* under exogenous GA_3_ and Flu treatments (Figure 6). While compared with the control, only *PsMYC2* was differentially expressed under exogenous hormone treatments, indicating its probable roles in hormone-mediated growth, development and the bud dormancy release of tree peony. *MYC* could mediate various hormone signaling pathways, and in *A. thaliana*, *AtMYC2* functions in ABA signaling pathways [6,30]. In *T. aestivum*, *TaMYC2*/4/5/6 were upregulated under both GA_3_ and ABA treatments and involved in plant development process [25]. In addition, we found that the expression level of *PsMYC2* was significantly upregulated under high-temperature stress (Figure 6). *MYCs* also have been reported to be involved in the plant abiotic stress response. In *T. aestivum*, *TaMYC4* and *TaMYC6* were upregulated under low temperatures and drought stress [25]. *BoMYCs* from *Brassica oleracea* plays an important role in response to sulfur stress [31].

However, in comparison to the control, the signal intensity of the *PsMYC2* promoter did not exhibit any significant differences when subjected to exogenous GA treatment (Figure 7). This suggests the presence of an indirect regulatory mechanism between the GA signal and expression regulation of *PsMYC2*. In *A. thaliana* and other plant species, the GA pathway has the ability to interact with various genetic pathways related to flowering or connect with other phytohormone pathways, thereby exerting a significant influence on plant growth [32]. This reveals that the GA pathway can regulate the plant development process through the GA-DELLA-MYC2/3 module [32,33,34,35], while the potential mechanism of the GA pathway in tree peony still needs further study. Furthermore, the interaction network prediction revealed that PsMYC2 not only formed heterodimers but also engaged in interactions with other proteins to influence the growth and development processes in tree peony (Figure 8). In *A. thaliana*, the discovery of *AtMYC2* as the first transcription factor regulated by JAZ proteins had been found to be involved in the defense regulation against insect herbivory [36]. This regulation occurrs in a partially redundant manner with its homologs *AtMYC3* and *AtMYC4* [4]. Furthermore, the proteins of MYC2, 3, 4, and 5 from *Isatis indigotica* have the ability to directly interact with MYB proteins, indicating redundant functions in response to Jasmonic acid [37]. The TIFY family plays a significant role in regulating various aspects of plant development, abiotic stresses, and phytohormone treatments [38]. In *Ananas comosus*, JAZ proteins have the capability to bind to MYC transcription factors and recruit TIFY proteins to regulate the growth and development process and abiotic stresses response [39]. These results indicate that *PsMYCs* might play vital roles in the growth and development regulation of tree peony, while the specific regulatory mechanisms still need to be further studied.

## 5. Conclusions

In this study, the identification of the MYC gene family in tree peony was conducted for the first time, followed by an analysis of gene structures, phylogenetic relationships, and expression profiles. Subsequently, the expression patterns of candidate genes *PsMYC1/2/9/10* were examined under various exogenous hormone and high-temperature treatments, revealing that only *PsMYC2* exhibited significant differences in expression levels. Furthermore, the promoter activity analysis of *PsMYC2* demonstrated its sensitivity to Flu signal and high-temperature stress. These findings provide a foundation for understanding the molecular mechanisms of *PsMYCs* in regulating the growth and development of tree peony.

## Figures and Tables

**Figure 1 plants-13-00437-f001:**
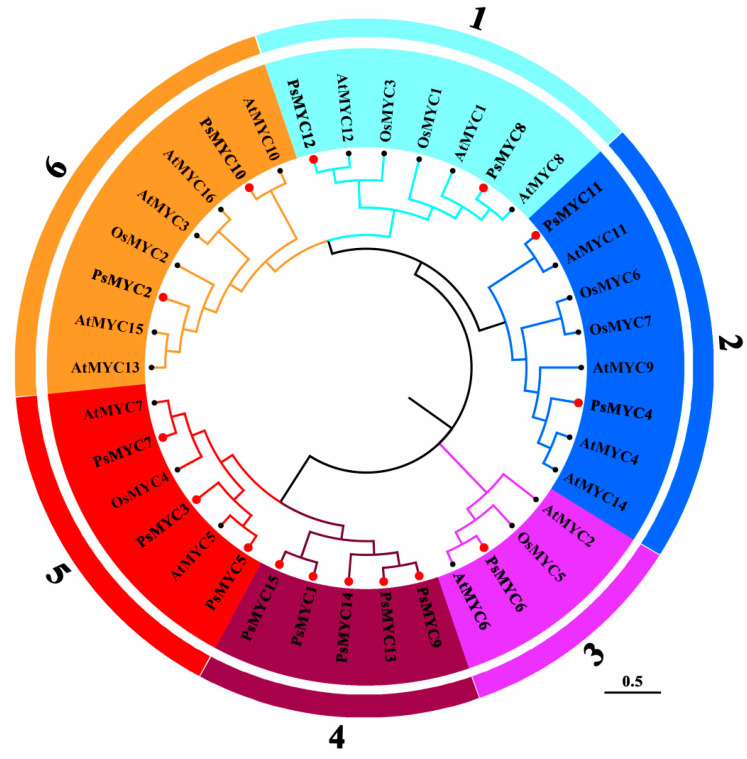
Phylogenetic relationship of PsMYC proteins. Different numbers and background colors represent six different groups of the MYC proteins named groups 1 to 6. The PsMYC proteins are marked by red points and others from *A. thaliana* and *O. sativa* were marked by black points. The At, Os, and Ps represent *A. thaliana*, *O. sativa*, and *P. suffruticosa*, respectively.

**Figure 2 plants-13-00437-f002:**
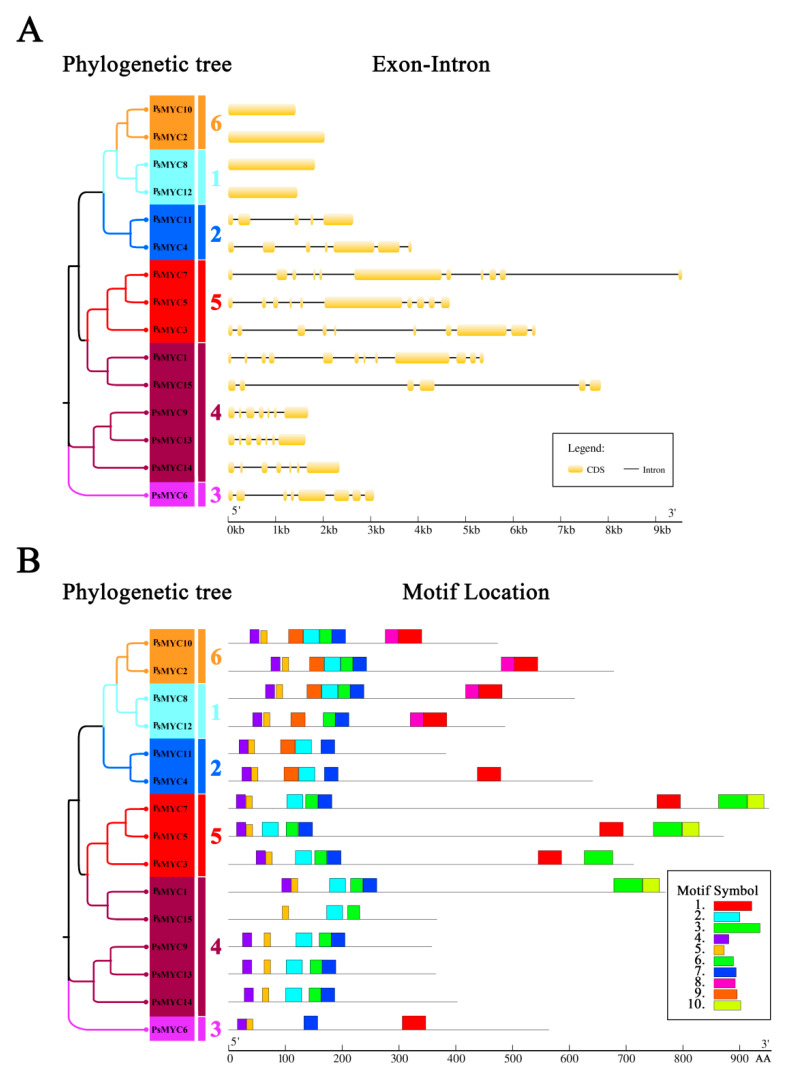
Gene structures and conserved motifs of the PsMYC proteins. (**A**) Gene structure analysis of the PsMYC proteins. In the phylogenetic tree, different colors represent different groups of the PsMYC proteins. In the exon–intron distribution, the black lines represent introns, and the yellow boxes represent the coding sequences. The horizontal value represents a gene length from 5′ to 3′. (**B**) Conserved motif analysis of the PsMYC proteins. In the phylogenetic tree, different colors represent different groups of the PsMYC proteins. In a motif location, different motif numbers are marked by different color boxes. The horizontal value represents a gene length from 5′ to 3′.

**Figure 3 plants-13-00437-f003:**
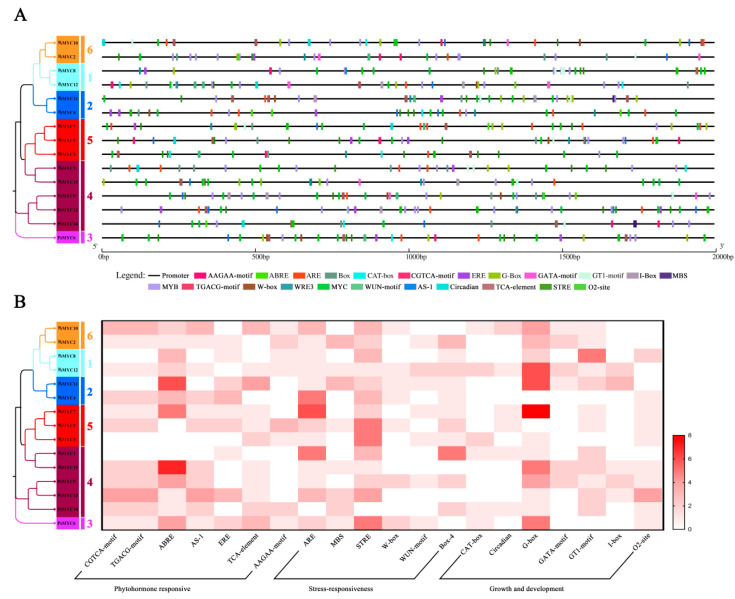
The *Cis*-regulatory element (CRE) analysis of *PsMYC* gene promoters. (**A**) The locations of CREs in the 15 *PsMYC* gene promoters. The black lines represent the 2 kb promoter regions, and the different color boxes correspond with the different kinds of CREs. The horizontal values represent the promoter length. (**B**) A heatmap of CREs in the 15 *PsMYC* gene promoters. The different boxes indicate the number of CREs in different *PsMYC* gene promoters. The white boxes represented no corresponding CRE, and the red boxes represent eight corresponding CREs. In the phylogenetic tree, different colors represent different groups of the PsMYC proteins.

**Figure 4 plants-13-00437-f004:**
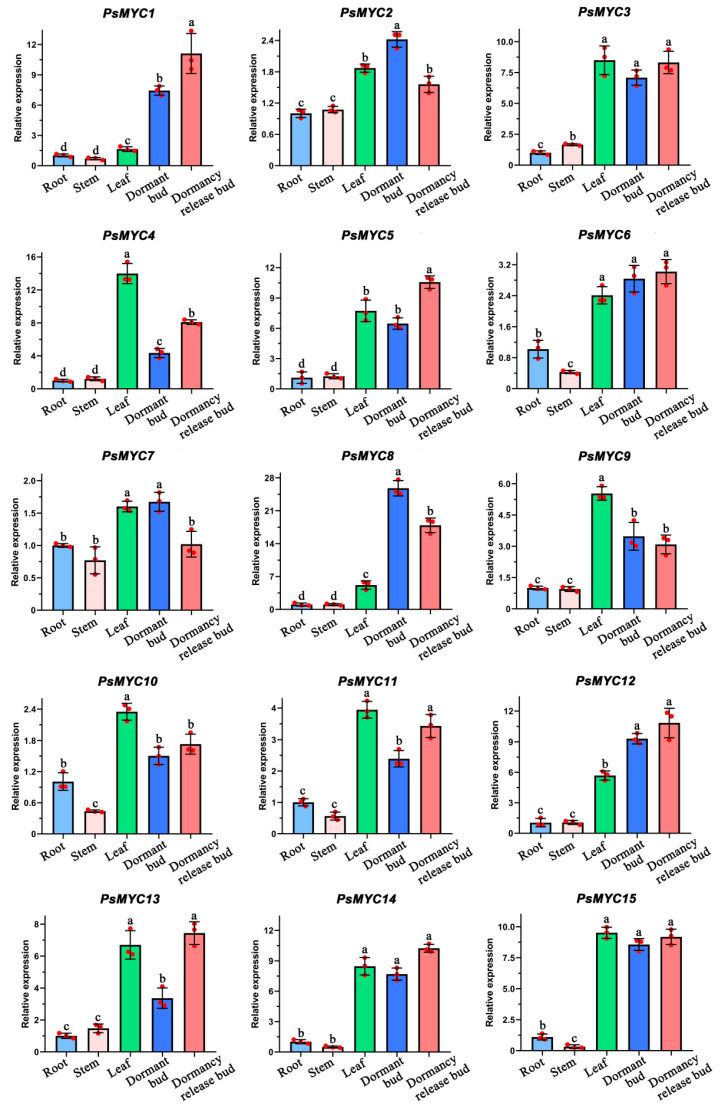
The expression patterns of *PsMYCs* in different tissues. Means ± SDs, *n* = 3, *p* < 0.05. Letters represent the expression significance of the *PsMYCs*.

**Figure 5 plants-13-00437-f005:**
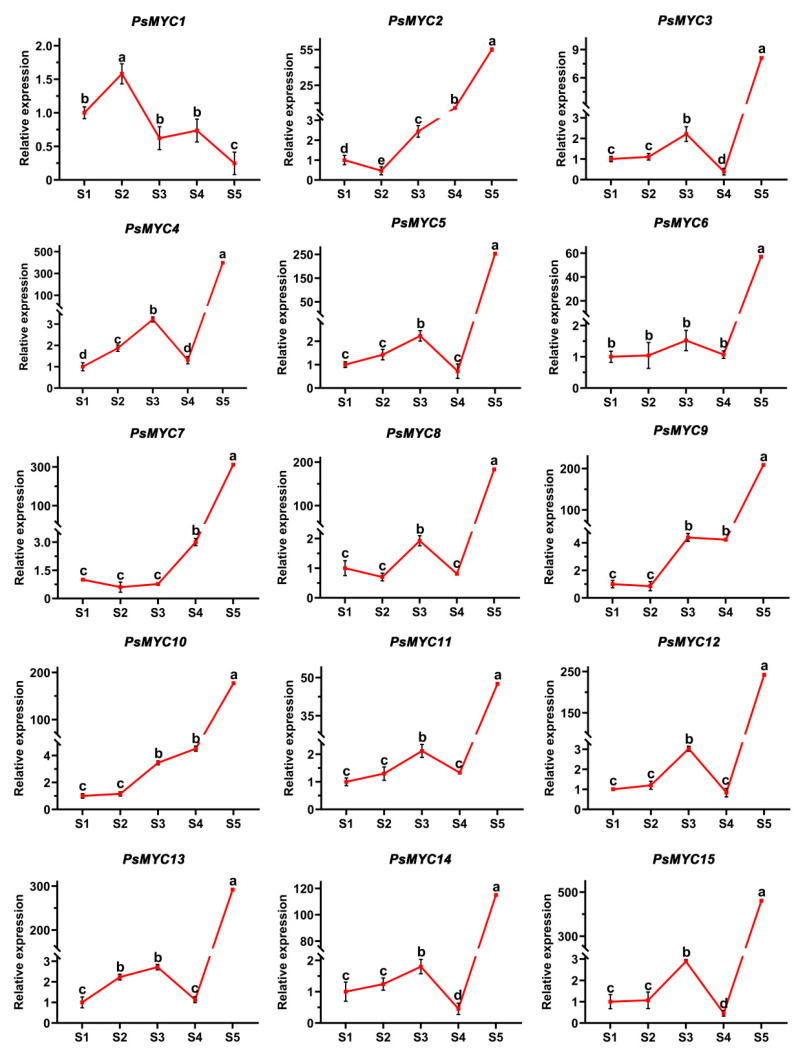
The expression patterns of *PsMYCs* in the growth and development process of tree peony. Means ± SDs, *n* = 3, *p* < 0.05. Letters represent the expression significance of the *PsMYCs*. S1–S5 represent different development stages. S1: bud dormancy stage; S2: bud dormancy release stage; S3: blooming bud stage; S4: wind bell stage; S5: full flowering stage.

**Figure 6 plants-13-00437-f006:**
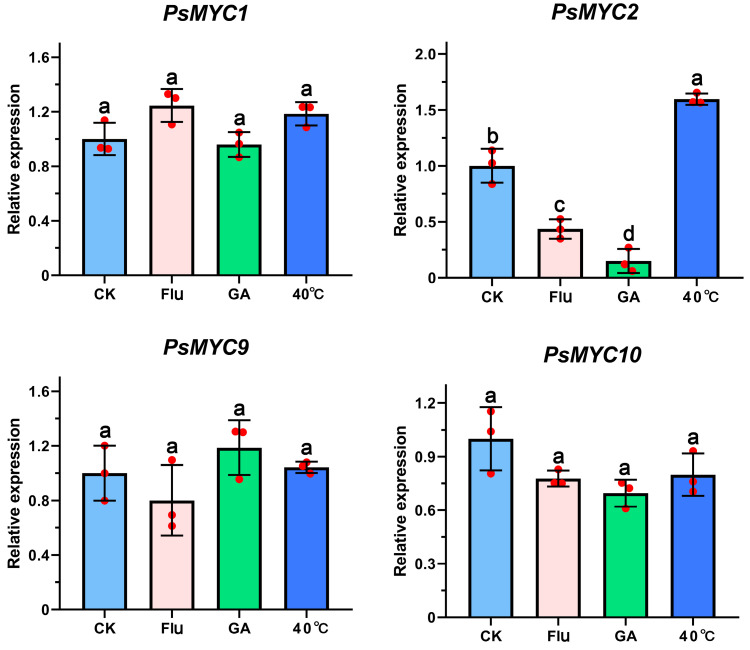
The expression patterns of *PsMYC1/2/9/10* under different treatments. Means ± SDs, *n* = 3, *p* < 0.05. Letters represent the expression significance of *PsMYCs*. CK: the control plants without treatment; Flu: the plants with 300 mg/L Flu treatment; GA: the plants with 300 mg/L GA_3_ treatment; 40 °C: the plants with 40 °C treatment.

**Figure 7 plants-13-00437-f007:**
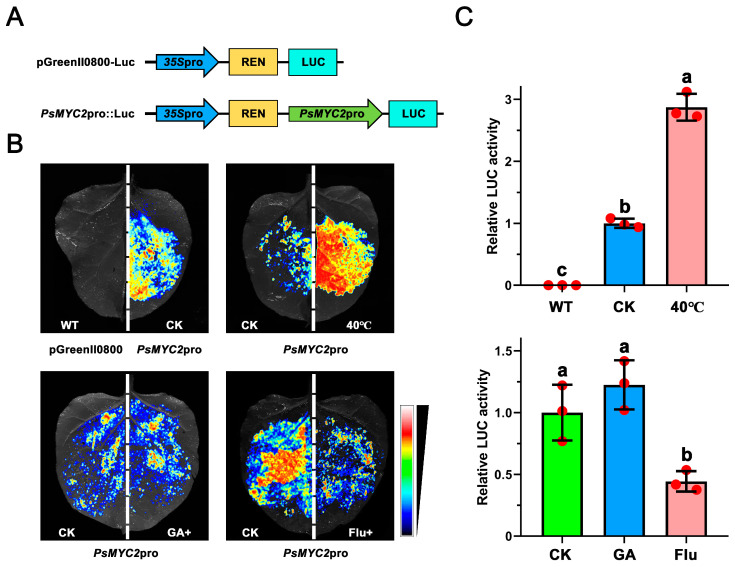
The promoter activity analysis of *PsMYC2* under different treatments. (**A**) The *PsMYC2*pro::Luc construct used as reporter plasmid for luciferase assays. (**B**) Luciferase assays via transient transformations into in *N. benthamiana* leaves with *PsMYC2*pro::Luc under different treatments. (**C**) Quantitative analysis of luciferase signal intensity under different temperature treatments. Means ± SDs, *n* = 3, *p* < 0.05. Letters represent the relative LUC activity significance under different treatments.

**Figure 8 plants-13-00437-f008:**
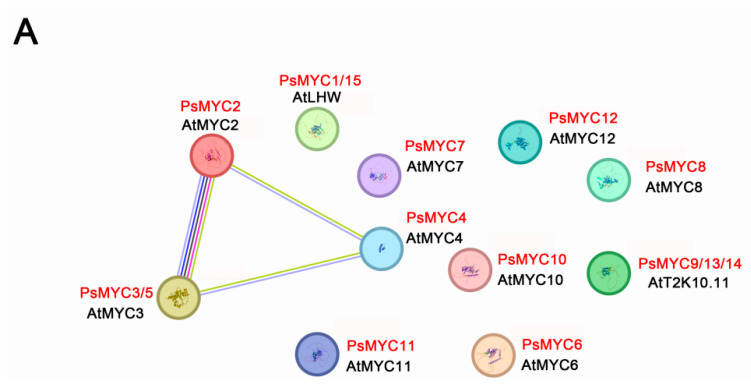
The prediction of the MYC protein interactions network. (**A**) Protein interaction network diagram of 15 PsMYC proteins. (**B**) Protein interaction network diagram of PsMYC2 proteins. Predicted protein interaction network map of PsMYC2 with other proteins in *A. thaliana*. Each node represents a corresponding protein, and the different colored lines indicate the type of evidence of protein interactions.

## Data Availability

The data presented in this study are available in Appendix A here.

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
