# Peer review of "Genome-Wide Identification of MYC Transcription Factors and Their Potential Functions in the Growth and Development Regulation of Tree Peony (Paeonia suffruticosa)"

_plants, 2024, doi:10.3390/plants13030437_

Round 1
Reviewer 1 Report
Comments and Suggestions for Authors
The MYC transcription factors, belonging to bHLH TF family, play a crucial role in regulation of development and stress responses in plant kingdom. This study systematically investigated the MYC TFs in tree peony genome and showed their expression profiles in various tissues, development stages, and under stress. The results indicated that the PsMYC2 TF may play an important role in growth and flowering transition in tree peony. This study may attract the attentions from tree peony research community for further investigation of molecular mechanisms in regulation of tree peony growth, flowering and stress response.
However, I require a revision to address my concerns regarding the mistakes, paper presentation, references, etc.
L9: It makes no sense to me: “a highly ornamental and economical woody plant”. I would revise this sentence to make it clear.
L10: I’m afraid the “However” at the beginning of this sentence is inappropriate.
L20: Please check through the manuscript to correct “cis-element” as “cis-element”.
L55: I would revise the sentence of “While the growth and …”.
L83-L85: Do you mean “All plant tissue materials” or “Total RNA samples” here?
L103: The Arabidopsis database is named as “TAIR”.
L104: The link for the rice plant database doesn’t work.
L139: I would provide the reference(s) for the 2−△△CT method.
L182: Please check across the manuscript to correct the font of the Latin name for some plant species.
L189: Are there previous studies showing the genome duplication events in the tree peony?
L191: Please check across the manuscript to correct the font for the gene/protein name.
L223: What is the meaning of the color key in Fig. 3B?
L246: I would define the developmental stages of S1-S5 here in the text.
L274: I would use the same symbol to show the difference levels in Fig. 4, 5 and 6.
L304: It makes no sense for the point why the authors showing the results presented in Fig 8B. Does it mean the PsMYC2 form the exact same protein interaction network as AtMYC2 since they share a high similarity in sequence, structure and functions?
Comments on the Quality of English LanguageI would revise and polish the manuscript to better present the results.
Author Response
Thanks for your comments on our paper, and we have revised our paper according to your comments. Please see the attachment.

Reviewer 2 Report
Comments and Suggestions for Authors
The manuscript with the title “Genome-wide identification of MYC transcription factors and their potential functions in growth and development regulation of Tree Peony (Paeonia suffruticosa)”, reports original findings on Myelocytomatosis transcriptional factors in tree peony.
Abstract provides a good summary of the manuscript.
Introduction highlights very well the state of the art on the topic and provides the adequate justification for the study.
Material and Method is divided in several subheadings describing the experimental assays in sufficient detail.
Results deals with gene structures, phylogenetic relationships, and expression profiles of the MYC gene family in tree peony. Results are consistent and overall have a good degree of novelty.
Conclusions are addressing the objectives proposed.
Out of 36 references, a good percentage are papers from recent years. The sources are on the topic.
Best regards.
Comments on the Quality of English LanguageEnglish style and syntax improvements are needed.
Author Response
Thanks for your comments on our paper, and according to your comments, we have improved the English style and syntax of the article.